# Regulatory T cells induce polarization of pro-repair macrophages by secreting sFGL2 into the endometriotic milieu

Xin-Xin Hou[1,2], Xiao-Qiu Wang[1,2], Wen-Jie Zhou[1] & Da-Jin Li [1✉]

An increased number of highly active regulatory T cells (Tregs) and macrophages has been found in peritoneal fluid from women with endometriosis. Here, we show that the level of Tregs-derived soluble fibrinogen-like protein 2 (sFGL2) increases in the peritoneal fluid of women with endometriosis. Higher expression of FGL2 and its receptor CD32B is observed in eutopic endometrium and ectopic tissues. The production of sFGL2 in Tregs may be enhanced by several cytokines. sFGL2 selectively induces pro-repair macrophage polarization mainly through the activation of the SHP2-ERK1/2-STAT3 signaling pathway, and the suppression of the NF-κB signaling pathway. Furthermore, sFGL2 induces a much higher level of metallothionein (MT) expression that in turn facilitates pro-repair macrophages polarization. sFGL2-induced pro-repair macrophages promote Th2 and Tregs differentiation, creating a positive feedback loop. These findings suggest that sFGL2 secreted by Tregs skews macrophages toward a pro-repair phenotype via SHP2-ERK1/2-STAT3 signaling pathway, which is involved in the progression of endometriosis.

[1] Laboratory for Reproductive Immunology, Key Laboratory of Reproduction Regulation of NPFPC, SIPPR, Shanghai Key Laboratory of Female Reproductive Endocrine Related Diseases, Hospital and Institute of Obstetrics and Gynecology, IRD, Fudan University Shanghai Medical College, Shanghai, China. [2] These authors contributed equally: Xin-Xin Hou, Xiao-Qiu Wang. ✉email: djli@shmu.edu.cn

Endometriosis is one of the most common disorders observed in fertile women and remains a major cause of pelvic pain and infertility worldwide[1]. Although it has been extensively studied, the precise etiology of endometriosis remains poorly understood. The pivotal role of immune regulation in endometriosis has been widely investigated, and a lack of adequate immune surveillance in the peritoneal cavity milieu is thought to be a cause of the disorder[2,3]. CD4+CD25+Foxp3+ Tregs are known to play an important role in controlling and modulating numerous immune responses[4]. Knowledge of the critical immune suppressive role of Tregs has helped us elucidate the pathogenesis of endometriosis. In cases of endometriosis, patients appear to have an increased number of highly active Tregs and elevated concentrations of suppressive molecules in the peritoneal fluid[5,6]. Moreover, Tregs densities observed within endometriotic lesions correlate to the stage and severity of endometriosis[7], suggesting that Tregs may be involved in the progression of endometriosis via immune suppression and in maintaining peritoneal cavity immune tolerance. However, the pathogenic mechanisms by which Tregs accomplish these tasks in vivo are still not well understood. Recent studies have highlighted the key concept that Tregs are functionally heterogeneous and that any given mechanism of immune suppression is tissue- and context-dependent[8,9]. Therefore, the relative contributions of Tregs-associated molecules to suppressive activity may depend on various factors including the genetic background of the host, the stimulant antigen, the experimental system and the site and type of the immune response. An improved understanding of the mechanisms by which Tregs mediate immune suppression in cases of endometriosis must be developed.

Fibrinogen-like protein 2 (FGL2) exists in both soluble and membrane forms. Soluble fibrinogen-like protein 2 (sFGL2) has been recently identified as a novel effector molecule of Tregs and plays a pivotal role in regulating both innate and adaptive immunity[10–12]. Previous microarray analysis has shown that Tregs have increased *FGL2* gene transcription levels[13]. Herman et al. were the first to report that abundant levels of FGL2 mRNA found in Tregs isolated from pancreas lesions may be involved in the prevention of experimental type 1 diabetes[14]. In subsequent studies, Fontenot et al. observed high levels of *FGL2* expression in Tregs isolated from wild-type Foxp3-GFP and IL-2-/- mice[13], while Gavin et al. confirmed that several genes could be amplified by FOXP3, including *FGL2*, *CD73*, *CD39*, and *CTLA-4*[15]. FGL2 has also been found to be co-expressed with FOXP3 in cardiac and liver allograft models, implying its role in tolerant liver and heart allografts[16]. Cumulatively, these observations support the notion that sFGL2 serves as an effector molecule of Tregs. FGL2 mediates its immunosuppressive activity by binding to inhibitory FcγRIIB (CD32B) receptors expressed by antigen presenting cells (APC), including dendritic cells (DC) and B cells, and thus sFGL2 may inhibit the maturation of DC, resulting in the suppression of effector T cell responses and inducing apoptosis of B cells[17,18]. However, nothing is known about sFGL2 and its potential roles in endometriosis.

As an essential effector cell of innate immunity, macrophages not only play a crucial role in immune defense but also in regulating inflammation, tissue repair, and remodeling[19,20]. Macrophages and their activation states are characterized by plasticity and flexibility. In the early 1990s, two different phenotypes of macrophages were described: one of them called classically activated (or inflammatory) macrophages (M1) and the other alternatively activated (or wound-healing) macrophages (M2). Currently, it is known that the functional polarization of macrophages into only two groups is an oversimplified description of macrophage heterogeneity and plasticity. The pro-inflammatory and pro-repair phenotypes of macrophages represent a dynamic and changing state of macrophages activation[21,22]. Pro-inflammatory macrophages release pro-inflammatory cytokines that inhibit the proliferation of surrounding cells and combat pathogens; unlike pro-inflammatory macrophages, pro-repair macrophages release anti-inflammatory cytokines that favor contiguous cell growth, survival, and tissue repair[23]. Macrophages polarization participates in many pathological progresses, such as cancer and inflammatory and autoimmune diseases[24]. An imbalance in macrophages polarization is often associated with various diseases and inflammatory conditions[25,26]. In endometriosis, previous studies have shown that macrophages display pro-inflammatory phenotypes in eutopic endometrium of women with endometriosis[27]. Further, a transcriptome meta-analysis of immune profile between eutopic endometrium from stage I–II to III–IV endometriosis suggests a predominant pro-inflammatory macrophage profile in eutopic endometrium from women with stage I–II endometriosis, while pro-repair macrophages are more prevalent in stage III–IV endometriosis[28]. Despite a series of studies exploring the molecular mechanisms of macrophages polarization, such mechanisms remain to be fully elucidated due to the complex network of signals associated with the cellular microenvironment.

Metallothionein (MT) is a group of small proteins and has been considered to serve as stress or acute phase proteins performing a variety of functions such as protection from oxidative damage, modulation of inflammation and angiogenesis[29–31]. Previous studies have demonstrated that MT can alter certain immune functions, and plays a key role in immunomodulation. MT has been found to have several immunosuppressive properties, such as suppressing cytotoxic T lymphocyte activity against allogeneic target cells[32] and suppressing natural killer cell (NK) activity[33]. And there is a higher expression of the MT2A isoform in IL-10-induced pro-repair macrophages[34]. However, the role of MT in contributing to modulate macrophage phenotype remains unclear.

Based on these observations and in view of the emerging significance of sFGL2 in Tregs-mediated immunosuppression, we postulate that Tregs-secreted sFGL2 contributes to the pathogenesis of endometriosis through multiple mechanisms. Herein, our data demonstrate that high levels of sFGL2 secreted by Tregs in endometriosis may shift macrophages polarization to the pro-repair phenotype through the suppression of NF-κB activity and the activation of the SHP2-ERK1/2-STAT3 pathway. We also observe a high MT expression in FGL2-treated macrophages, which furthermore can contribute to a pro-repair macrophage phenotype, in line with the immunosuppressive role of MT evidenced. Our study may prove beneficial in improving our understanding of the immune microenvironment in endometriosis.

## Results

**Higher levels of FGL2 in peritoneal fluid and endometriotic tissues in patients with endometriosis.** To investigate the roles of FGL2 in endometriosis, peritoneal fluid and endometriotic tissues were collected and analyzed. Higher levels of sFGL2 were detected in the peritoneal fluid of women with endometriosis compared to those without endometriosis, and sFGL2 levels were even higher in stage III–IV endometriosis (Fig. 1a). Consistent with higher sFGL2 concentrations found in peritoneal fluid of patients with endometriosis, FGL2 expression in eutopic endometrium increased compared to that of the normal endometrium, and even higher expression levels were found in endometriotic lesions (ovarian endometriosis) (Fig. 1b). An IHC analysis showed that CD32B (the receptor of FGL2) exhibited the highest levels of expression in ectopic lesions (ovarian endometriosis), lower expression levels in eutopic endometrium, and the lowest expression levels in the normal endometrium (Fig. 1c).

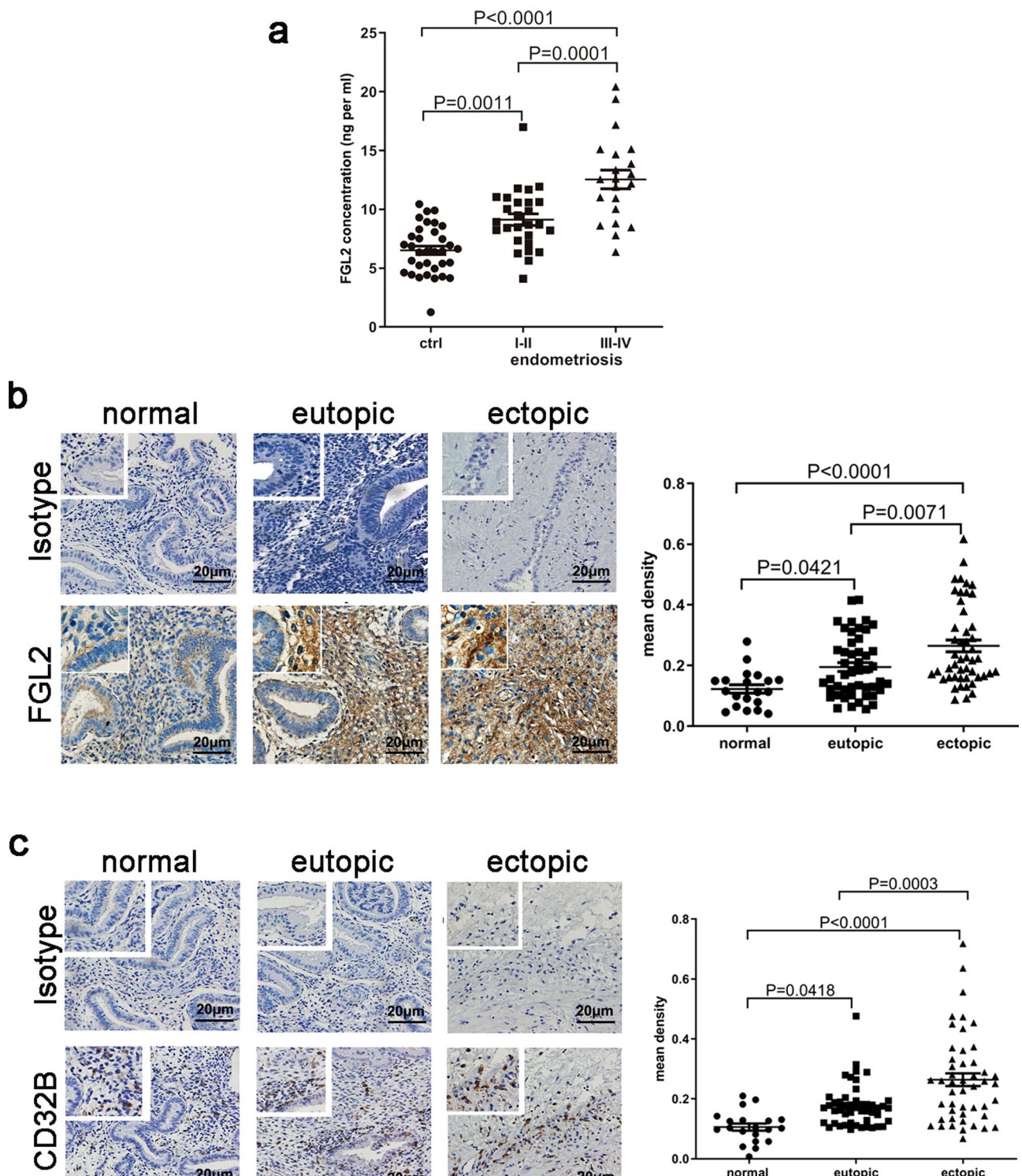

**Fig. 1 FGL2 levels in endometrium, ectopic tissues, and peritoneal fluid of women with or without endometriosis. a** Concentrations of sFGL2 in peritoneal fluid of women with (stage I–II, stage III–IV) or without endometriosis. **b** Immunohistochemistry staining of FGL2 in normal, eutopic endometrium and ectopic lesions; the density was analyzed using Image Pro Plus 6.0 software. **c** Immunohistochemistry staining of CD32B in normal, eutopic endometrium and ectopic tissues; densities were analyzed using Image Pro Plus 6.0 software. Data represent the mean value ± SEM.

**The expression of FGL2 and its receptor CD32B in endometrium and immune cells.** To better understand the source of sFGL2, we measured sFGL2 concentrations in cell culture supernatant of endometrial stromal cells (ESCs) and endometrial epithelial cells (EECs) (primary culture of ESCs and EECs from women without endometriosis) as well as monocyte, macrophages,

naive T cells, NK cells, Teffs and Tregs (peripheral venous blood samples from healthy volunteers) by using FGL2 ELISA kits. We also determined mRNA levels of FGL2 by using qRT-PCR. As shown in Fig. 2, the secretion of sFGL2 was mainly found in Tregs and naive T cells (Fig. 2a). FGL2 qRT-PCR also showed the highest levels of expression in Tregs (Fig. 2b). We also detected

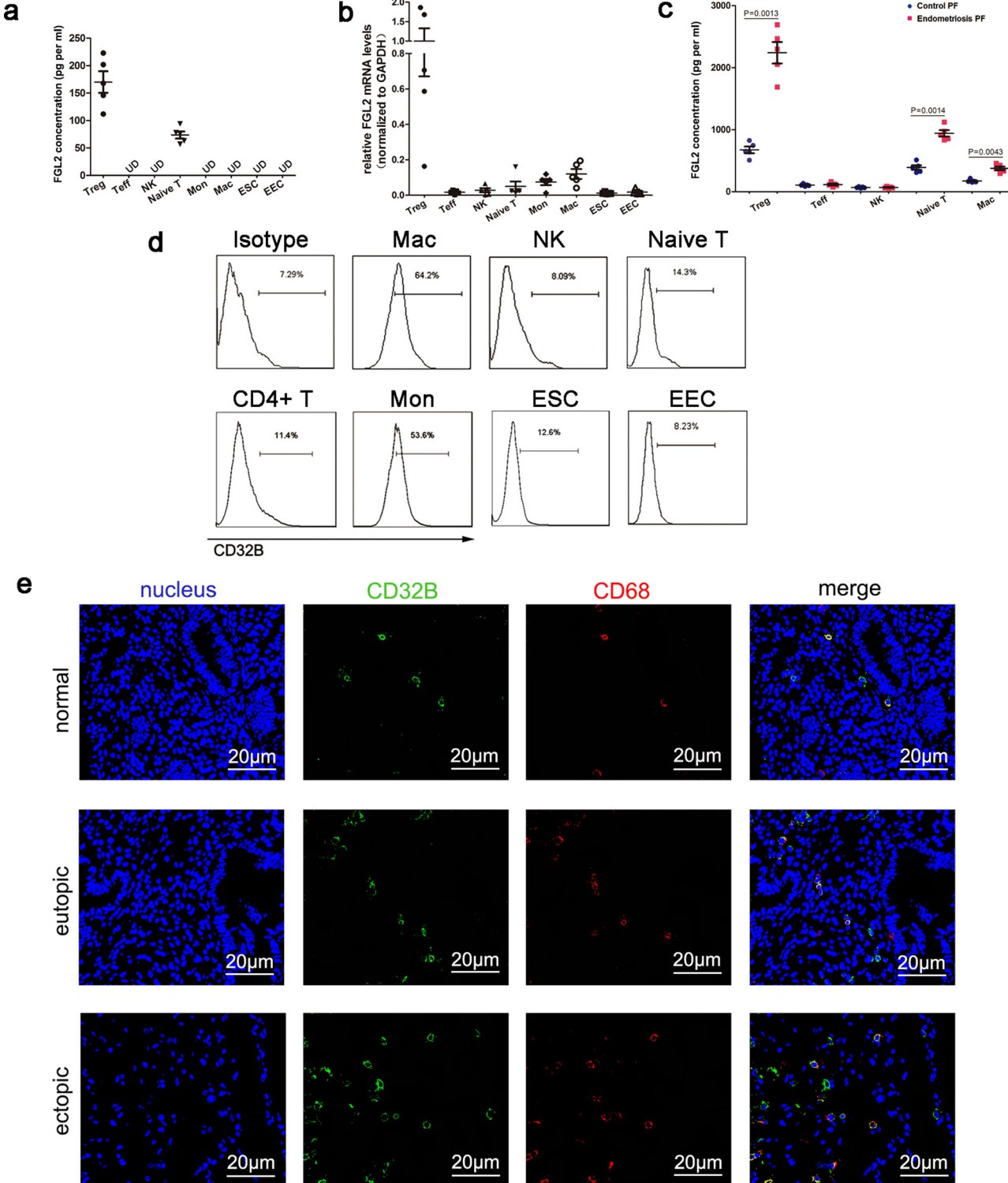

**Fig. 2 The expression of FGL2 and CD32B on isolated cells and in tissues. a** ELISA for FGL2 concentrations in cell culture supernatant. **b** FGL2 mRNA expression levels in different kinds of cells. (**a**, **b**) The individual immune cell populations were isolated from PBMC. EECs and ESCs from endometrium were primary cultured. **c** ELISA for FGL2 concentrations in cell culture supernatant. The individual immune cell populations were isolated from the peritoneal fluid of women with or without endometriosis (PF: peritoneal fluid). **d** CD32B expression on cell surfaces was measured by FCM. **e** Confocal microscopic analysis of nucleus (blue), CD32B (green), CD68 (red), expression in normal endometrium, eutopic endometrium, and ectopic lesions. Data represent the mean value ± SEM.

FGL2 levels of individual immune cell populations from peritoneal fluid of women with or without endometriosis and we found that Tregs in peritoneal fluid of endometriosis patients secreted significantly higher levels of FGL2 than that of control (Fig. 2c). Therefore, high levels of sFGL2 in peritoneal fluid of patients with

endometriosis were mainly produced by Tregs. CD32B was mainly expressed on monocyte and macrophages, and showed lower levels of expression in other cells (Fig. 2d). Confocal microscopy showed that the number of CD32B positive cells increased in both eutopic endometrium and ectopic tissues (ovarian endometriosis)

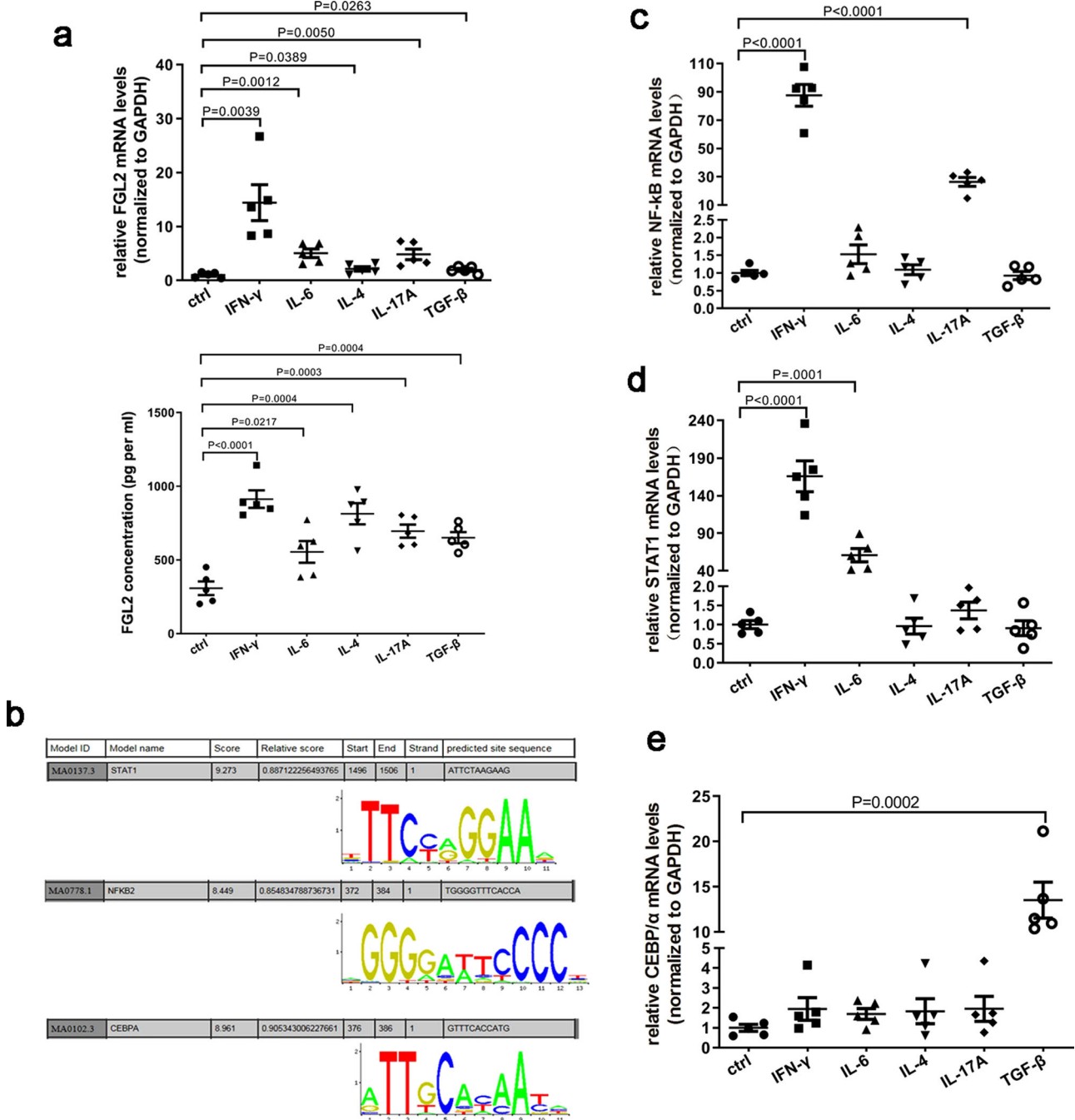

**Fig. 3 Cytokines induced FGL2 expression via different pathways in Tregs. a** Tregs were isolated and stimulated with indicated cytokines. FGL2 mRNA was examined by qRT-PCR after 24 h, the concentration of FGL2 in the supernatant was detected by ELISA. **b** Predicted binding sites of transcription factors STAT1, NF-κB, and C/EBPα on the FGL2 promoter region. **c–e** Differentially expressed NF-κB (**c**), STAT1 (**d**), and C/EBPα (**e**) genes in Tregs treated with indicated cytokines. Data represent the mean value ± SEM of five individual experiments ($n = 5$).

compared to the normal endometrium where ectopic lesions presented the highest expression levels (Fig. 2e and Supplementary Fig. 4a, b). Consistent with Fig. 2d, CD32B positive cells were mainly macrophages (CD68 is highly expressed by human macrophages) (Fig. 2e), and a few were NK and T cells (Supplementary Fig. 4a, b).

**Several cytokines induce FGL2 expression through different signal pathways.** In peritoneal cavities of women with endometriosis, concentrations of multiple cytokines increase, and specific factors (such as viral infections and cytokines) can induce high

levels of FGL2. We thus tested FGL2 expression in Tregs treated with different cytokines. We found that Tregs expressed higher levels of FGL2 to different extents after treatment with IFN-γ, IL-6, IL-4, IL-17A, and TGF-β (Fig. 3a). To better understand how these cytokines induced FGL2 expression, we searched the genomic region of FGL2 for transcription factor binding sites. Using the Jaspar database, we found that FGL2 contains binding sites for transcription factors NF-κB, STAT1 and C/EBPα (Fig. 3b). We thus examined the expression of these transcription factors and found that these transcription factors were differently regulated after treatment with several types of cytokines (Fig. 3c–e). We

speculate the mechanism by which IFN-γ may induce FGL2 expression through NF-κB and STAT1 pathways (Fig. 3c, d), IL-6 induces FGL2 expression mainly through STAT1 (Fig. 3d), IL-17A induces FGL2 expression mainly through NF-κB (Fig. 3c), and TGF-β induces FGL2 expression mainly through C/EBPα (Fig. 3e). However, IL-4 had no obvious effects on the expression of NF-κB, STAT1, and C/EBPα. We speculate that IL-4-induced FGL2 expression may occur through other signal pathways.

**sFGL2 induces pro-repair macrophages skewing by binding to CD32B (FcγRIIB).** To understand whether sFGL2 induces pro-repair macrophages polarization, we used an HTA 2.0 microarray to detect the expression profiles of genes that may exhibit dynamic polarization in macrophages treated with recombinant human FGL2 (rFGL2). We identified 1582 genes that were differentially expressed between macrophages treated with or without rFGL2 (Fig. 4a). Annotated GO biological processes were assigned to the differentially expressed genes, and the main biological processes observed were highly associated with the immune response and chemotaxis, and notably with other processes including cellular responses to Zinc ions, apoptotic processes, and small molecule metabolic processes (Fig. 4b). As was expected, macrophages treated with rFGL2 preferentially expressed tissue-protective and wound healing genes such as *CD163*, *IL-10*, *ARG1*, *FGF2*, *TGF-β1*, and *MMP9* and presented decreased levels of costimulatory molecules such as *CD80* and *CD86*. Overall, the rFGL2-treated macrophages resembled alternative pro-repair macrophages (Fig. 4c). To verify the microarray results, we used monocytic cell lines THP-1 and U937 cells for further studies. For differentiation to macrophages, THP-1 and U937 cells were treated with 100 ng per ml phorbol 12-myristate 13-acetate (PMA) for 48 h. Changes in cell morphology with increased adhesion and extended pseudopodia indicated the occurrence of differentiation (Supplementary Fig. 1). Similarly, treatment with rFGL2 efficiently induced pro-repair macrophages shifts in THP-1 and U937 cells (Fig. 4d, e).

CD32B, the receptor of FGL2, is expressed on macrophages. As macrophages are the most common cells found in peritoneal cavities in patients with endometriosis, we explored the immunoregulatory role of sFGL2 in macrophage polarization. To investigate the effects of sFGL2 on macrophages, we treated macrophages with rFGL2 or with an anti-CD32B neutralizing antibody, and then measured changes in macrophages phenotype and cytokine production. Notably, the macrophages treated with rFGL2 showed decreased levels of CD86 and HLA-DR expression and of pro-inflammatory cytokines IL-6 and IL-12p70 production (Fig. 4f). The macrophages treated with rFGL2 expressed more pro-repair markers CD163 and CD206, which was accompanied by an increase in the levels of the anti-inflammatory cytokines TGF-β and IL-10 (Fig. 4g), and the anti-CD32B neutralizing antibody pretreatment before adding rFGL2 efficiently blocked such effects (Fig. 4f, g). These data suggest that sFGL2-induced pro-repair macrophages polarization is CD32B-dependent.

**sFGL2 secreted by Tregs polarized macrophages to the pro-repair phenotype.** Macrophages were indirectly co-cultured with Tregs, rFGL2, or anti-FGL2 antibody were added separately. The macrophages co-cultured with Tregs showed decreased levels of CD86, HLA-DR expression (Fig. 5a) and IL-6, IL-12p70 production (Fig. 5b). The macrophages co-cultured with Tregs also showed increased levels of CD163, CD206 expression (Fig. 5c) and TGF-β, IL-10 production (Fig. 5d). rFGL2 further enhanced these effects, while anti-FGL2 antibody partly blocked these effects (Fig. 5a–d).

**sFGL2 induces pro-repair macrophages phenotypes through suppression of NF-κB and activation of the SHP2-ERK1/2-STAT3 pathway.** We next examined signaling pathways and transcription factors that directly affect macrophages polarization upon treatment with rFGL2. Our gene expression microarray analysis showed that the gene expression profiles undergo profound changes in signaling molecules associated with macrophages polarization, and the pathway network construction indicated that MAPK and NF-κB cascades were the two main signaling pathways altered in macrophages treated with rFGL2. We further performed a western blot to identify changes in core elements observed in the MAPK and NF-κB pathways in THP-1 and U937 cells. As is shown in Fig. 6a, rFGL2 induced the phosphorylation of ERK1/2 and decreased the phosphorylation of NF-κB, while p38 and JNK remained unchanged (Supplementary Fig. 2a). As a predominance of STAT3 and STAT6 activation promotes pro-repair macrophages polarization, we observed a stronger phosphorylation of STAT3, but not of STAT6, in cells treated with rFGL2 (Fig. 6a and Supplementary Fig. 2a).

CD32B contains an immuno-receptor tyrosine-based inhibitory motif (ITIM) in its cytoplasmic tail, and the inhibitory actions of ITIM-containing receptors have been linked to the recruitment of proteins such as SH2-containing tyrosine phosphatases (SHP) SHP1 and SHP2, and the SH2 domain contains inositol phosphatases (SHIP)[35,36]. To identify whether the role of rFGL2 is mediated by the ITIM, we next examined SHP1, SHP2, and SHIP activity. Notably, rFGL2-treated macrophages exhibited enhanced SHP2 activity (Fig. 6b), but not enhanced activity in SHP1 and SHIP (Supplementary Fig. 2b). To explore whether the changes in cellular signal pathways observed were caused by SHP2 phosphorylation, we used SHP2 and ERK inhibitors for further study. We found that ERK activation was inhibited in macrophages treated with an SHP2 inhibitor (Fig. 6c), which was accompanied by a reduced activation of STAT3 in macrophages treated with SHP2 or ERK inhibitors (Fig. 6c). However, neither inhibitor altered NF-κB activation (Supplementary Fig. 2c). Collectively, these data suggest that FGL2-induced pro-repair macrophages polarization may mainly occur through the activation of the SHP2-ERK1/2-STAT3 pathway and the repression of the NF-κB pathway.

**FGL2 induces MT expression in macrophages through STAT3 activation.** The microarray results revealed the occurrence of strongly upregulated MT expression in macrophages after rFGL2 treatment (Fig. 7a). To confirm our observations in the microarray results, we examined the effects of rFGL2 on MT expression using qRT-PCR. As is indicated in Fig. 7b, rFGL2-treated MDM significantly increased MT expression with MT2A and MT1F as the predominant upregulated isoforms. Therefore, we selected MT1F and MT2A as our main research objects in the following study. rFGL2-treated THP-1 and U937 cells also showed considerable induction of MT1F and MT2A expression (Fig. 7c). Given previous reports that have identified STAT binding sites on MT promotor[37], we hypothesized that STAT3 may directly regulate MT expression. We thus examined MT expression in macrophages treated with a STAT3 chemical inhibitor, and as expected the inhibition of STAT3 reduced MT1F and MT2A expression (Fig. 7d). These data indicate that the expression of MT is regulated by FGL2 via STAT3 signaling pathway.

**High levels of MT may promote pro-repair macrophages polarization.** To confirm the critical roles of MT in macrophages polarization, we investigated MT expression levels in differently polarized macrophages. We found that MT expression was, especially MT2A and MT1F, extremely high in pro-repair

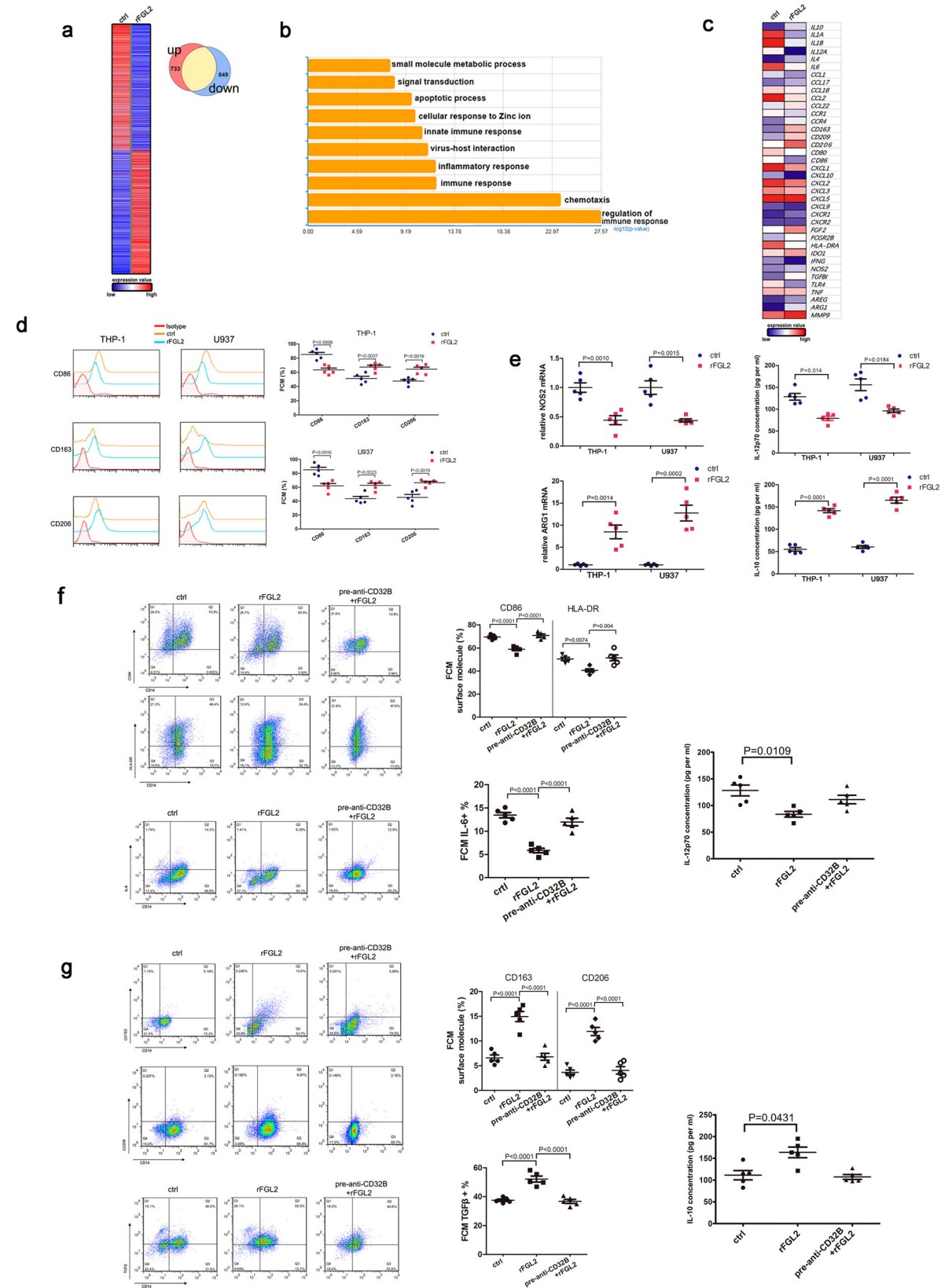

macrophages (Fig. 8a). MT (MT2A and MT1F) analysis in THP-1 and U937 cells showed similar findings to those obtained for MDM (Fig. 8b). Therefore, we postulated that macrophages producing high levels of MT may exhibit a pro-repair phenotype. We further transfected MT1F and MT2A expression vectors into

THP-1 and U937 cells (Supplementary Fig. 3) and investigated the phenotypes of these cells. As expected, MT1F and MT2A vector-transfected cells exhibited a pro-repair phenotype with reduced levels of the costimulatory molecule CD86 and increased levels of CD163 and CD206 (Fig. 8c). In addition, NOS2 mRNA

**Fig. 4 FGL2 induced pro-repair macrophages polarization through binding to CD32B. a** Heat map of differentially expressed genes in macrophages treated with PBS (crtl) or rFGL2. **b** Enriched GO analysis of the top ten GO terms changed in macrophages treated with rFGL2. **c** Heat map of representative pro-inflammatory and pro-repair related genes differentially expressed by macrophages treated with rFGL2. D-E, Flow cytometric analysis (**d**), qRT-PCR and ELISA analysis (**e**) for the expression of pro-inflammatory (CD86, NOS2, IL-12p70) and pro-repair (CD163, CD206, Arg1, IL-10) macrophages markers in THP-1 and U937 cells treated with rFGL2 or PBS. **f** CD86, HLA-DR, IL-6, and IL-12p70 expression in macrophages treated with rFGL2 or pretreated with anti-CD32B antibody 2 h before adding rFGL2 (0.5 µg per ml). **g** CD163, CD206, TGF-β, and IL-10 expression in macrophages treated with rFGL2 or pretreated with anti-CD32B antibody 2 h before adding rFGL2 (0.5 µg per ml). Data represent the mean value ± SEM of five individual experiments (n = 5).

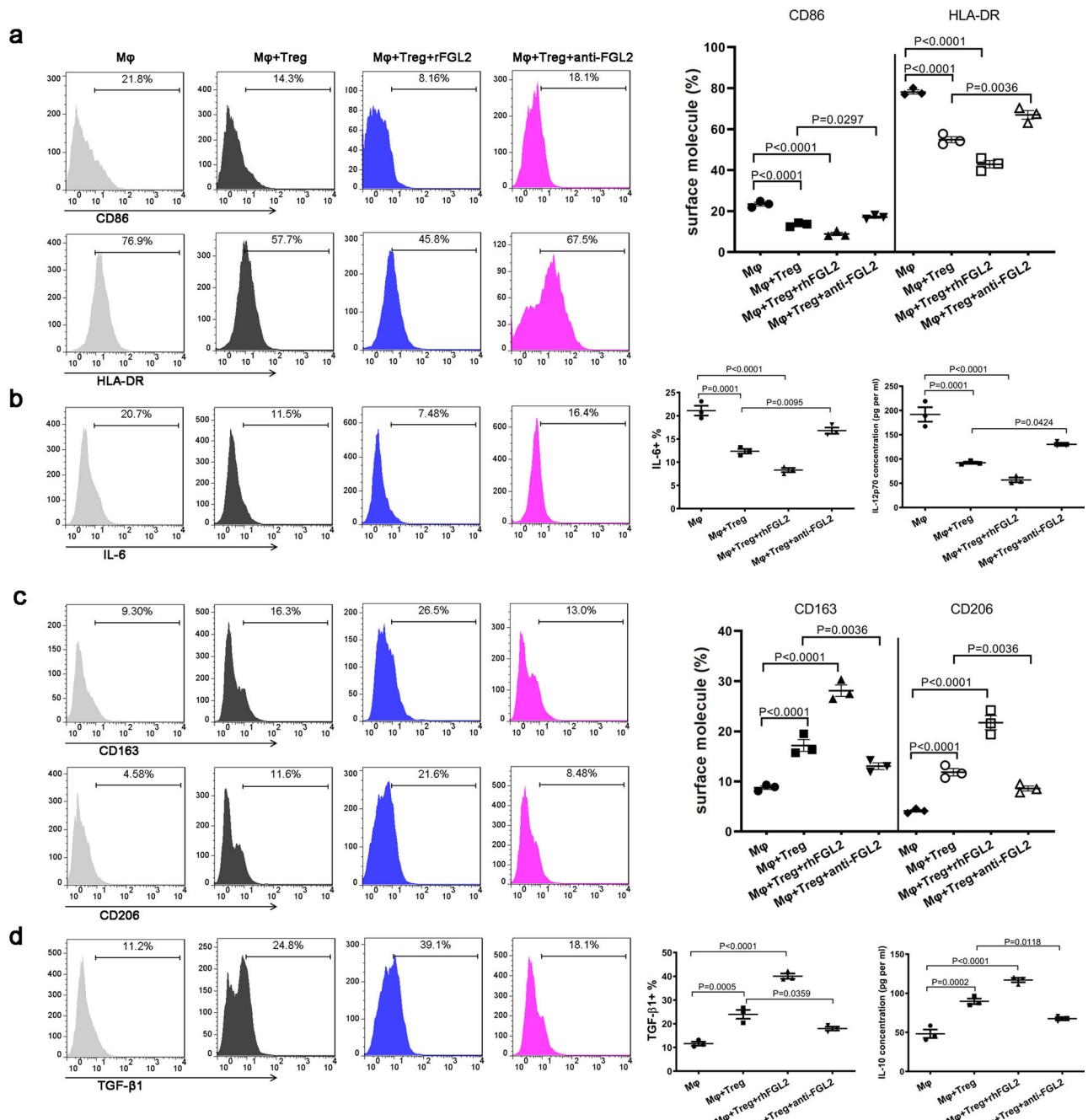

**Fig. 5 Tregs promoted pro-repair macrophages polarization via secreting sFGL2.** Macrophages (MΦ) were indirectly co-cultured with Tregs by transwell, recombinant human FGL2 (rhFGL2), or anti-FGL2 antibody were added separately. The expression of CD86, HLA-DR, CD163, CD206, IL-6, TGF-β1 were analyzed by FCM (**a–d**). The production of IL-12p70 and IL-10 were detected by ELISA (**b, d**). Data represent the mean value ± SEM of three individual experiments (n = 3).

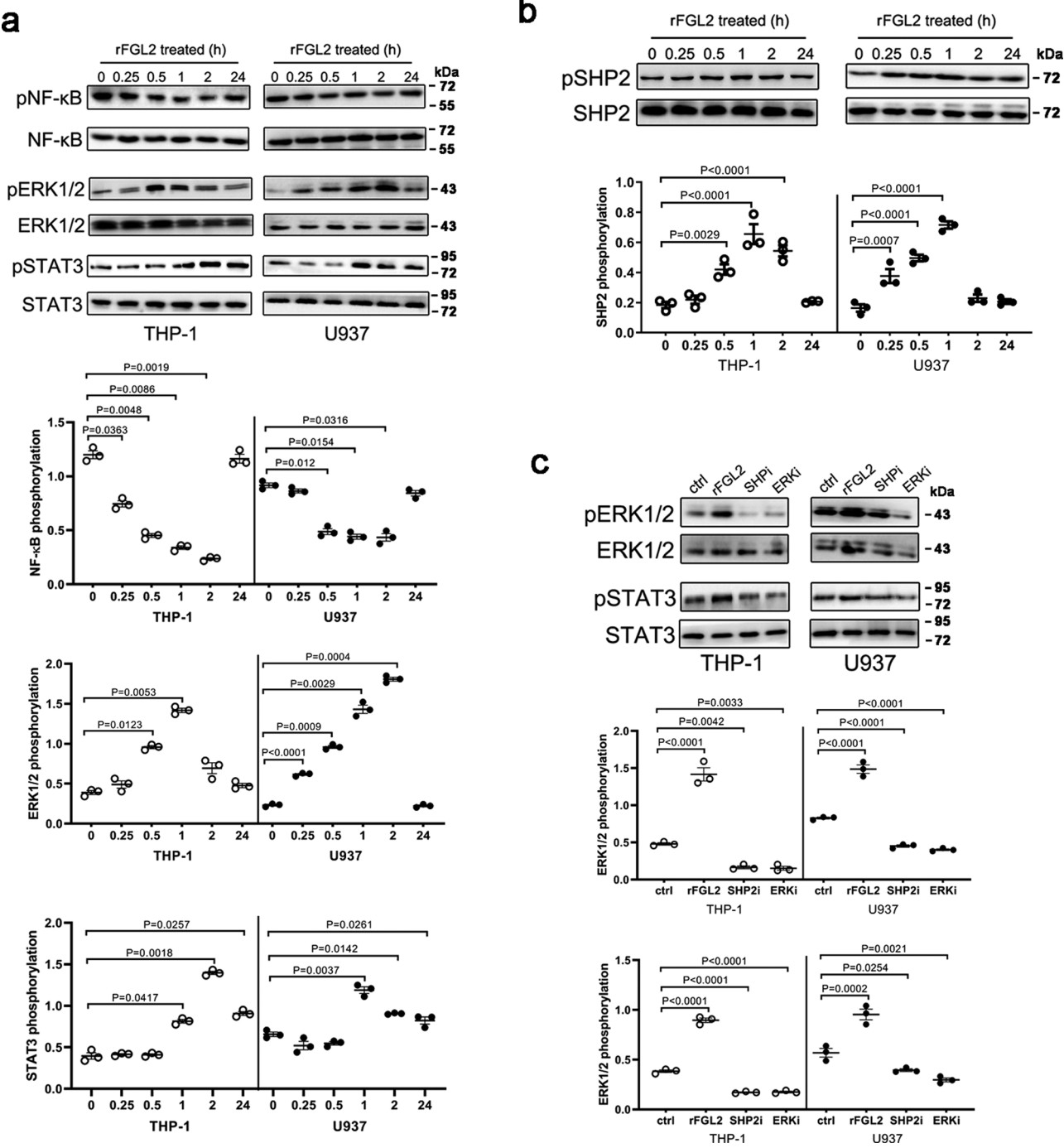

**Fig. 6 FGL2 induced pro-repair macrophages polarization through the suppression of the NF-κB signaling pathway and the activation of the SHP2-ERK1/2-STAT3 signaling pathway.** THP-1 and U937 cells were treated with rFGL2 (**a**, **b**) or SHPi/ERKi (**c**). The phosphorylation levels of NF-κB, ERK1/2, STAT3, and SHP2 in THP-1 and U937 cells were determined by western blot (**a–c**). Data represent the mean value ± SEM of three individual experiments (n = 3).

levels were reduced while ARG1 mRNA levels were increased in the MT1F and MT2A vector-transfected cells (Fig. 8d). We next explored whether MT deficiencies might also affect macrophages polarization. We used MT1F and MT2A siRNA to silence MT expression (Supplementary Fig. 3). However, with the exception of decreased levels of NOS2, we observed equivalent expression levels of CD86, CD163 and CD206 in MT1F and MT2A silenced macrophages compared to the control (Fig. 8e, f). Together, these data establish that proper MT expression is critical for macrophages functions and that abnormal inflammatory

responses mediated by macrophages may be associated with the aberrant expression of MT.

**FGL2-induced pro-repair macrophages promote Th2 and Tregs differentiation.** FGL2-treated macrophages were cocultured with naive T cells, and the mRNA levels of *TBX21* and *RORC* in T cells decreased, while the mRNA levels of *GATA3* and *FOXP3* increased (Fig. 9a). FGL2-treated macrophages reduced naive T cells secreting IFN-γ and IL-17A, but elevated naive

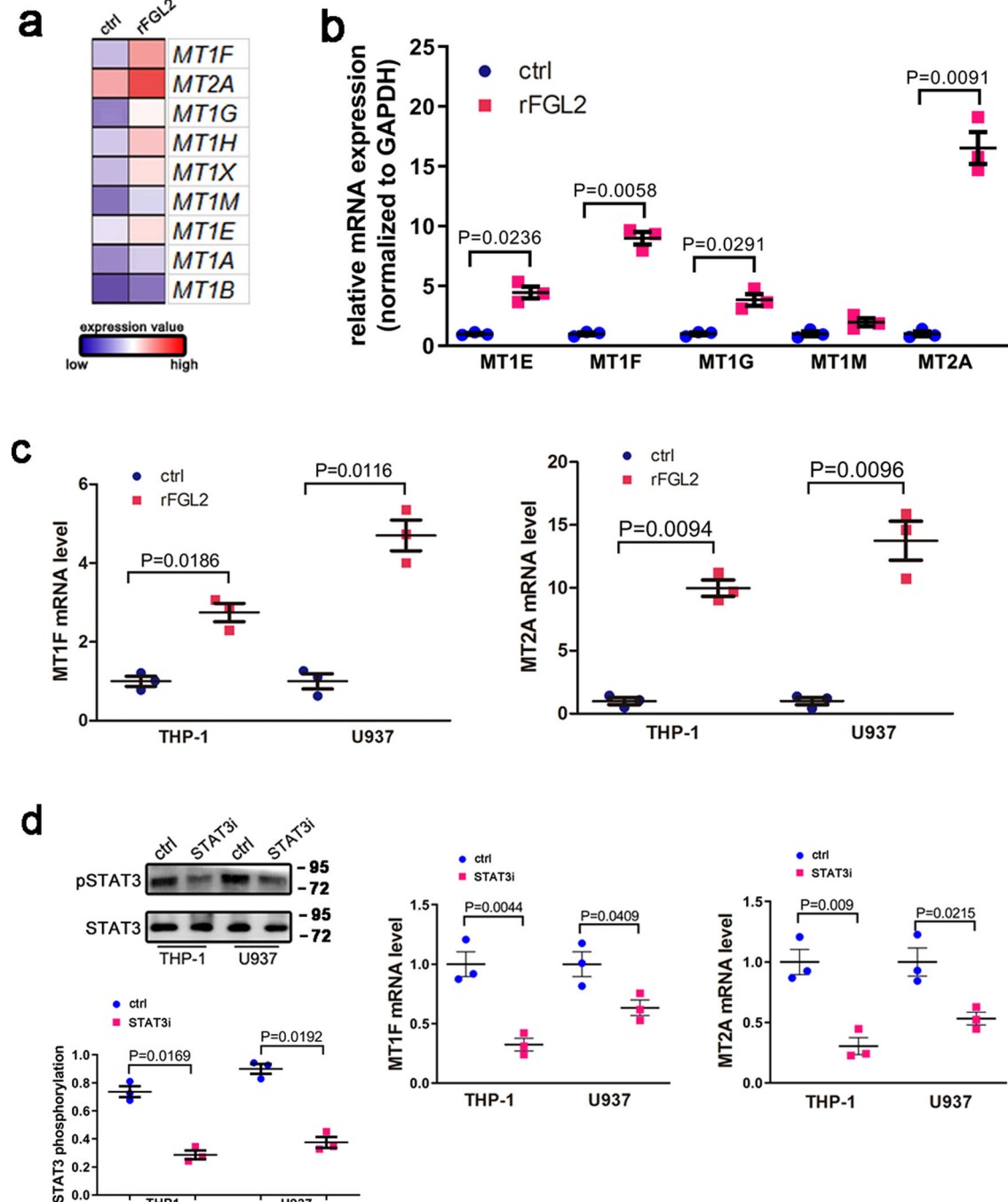

**Fig. 7 FGL2 increased MT expression in macrophages via activation of STAT3 signaling pathway. a** Heat map of MT expression in macrophages treated with rFGL2. **b** qRT-PCR-verified MT expression in rFGL2-treated MDM. **c** Selected MT (MT1F and MT2A) expression analysis in THP-1 and U937 cells treated with rFGL2. **d** STAT3 was responsible for MT expression; THP-1 and U937 cells treated with STAT3 inhibitor expressed lower levels of MT1F and MT2A. Data represent the mean value ± SEM of three individual experiments ($n = 3$).

T cells secreting IL-4 (Fig. 9b). A schematic diagram of these results is presented in Fig. 10.

## Discussion

The identification of novel Tregs effector molecules and the subsequent recognition of their diverse functions have provided new insights into how Tregs control local immune responses and serve as attractive therapeutic targets. FGL2 has been recently reported as a Tregs effector molecule that suppresses Th1, but enhances Th2 responses, and downregulates antigen presentation

activities of APC through the CD32B pathway[17]. While the importance of FGL2 for numerous diseases has been established, much less is known about the role of FGL2 in endometriosis. The current study focused on identifying whether FGL2 is a key effector molecule of Tregs involved in the pathogenesis of endometriosis.

The data derived from our study reveal that the expression of FGL2 in ectopic tissues and eutopic endometrium of women with endometriosis is much higher than that of the endometrium of women without the disease and varies with the stage of the disease, suggesting a potential role of FGL2 in progression of this

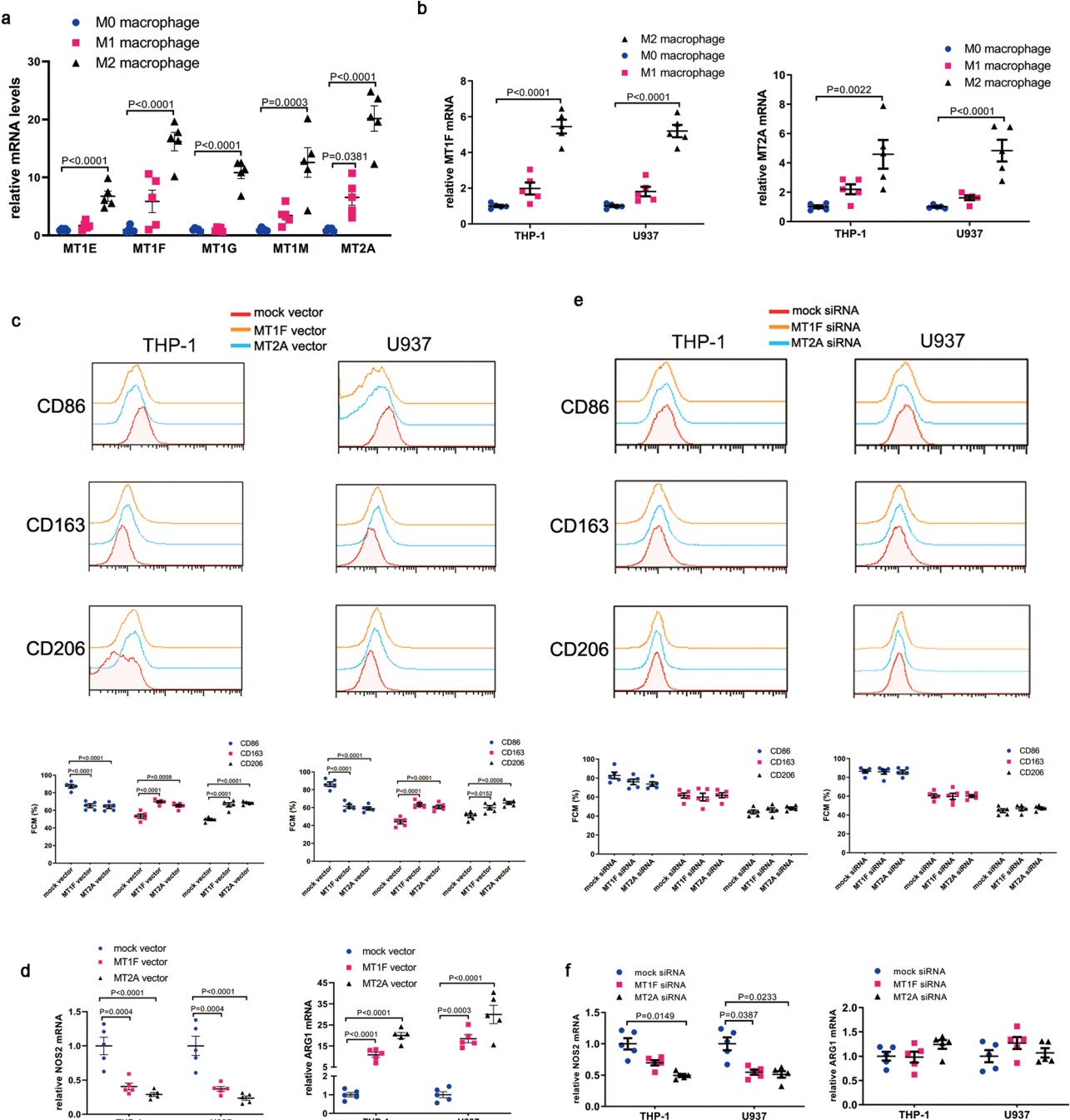

**Fig. 8 Higher MT expression promoted pro-repair macrophages polarization. a** MT expression is differentially polarized MDM. **b** MT1F and MT2A expression in differentially polarized THP-1 and U937 cells. **c**, **d** The overexpression of MT1F and MT2A in THP-1 and U937 cells induced pro-repair macrophages polarization. **e**, **f** The knockdown of MT1F and MT2A in THP-1 and U937 cells had no effect on macrophage polarization. Data represent the mean value ± SEM of five individual experiments ($n = 5$).

disease. However, the normal endometrium of the controls was obtained from patients who had symptoms of abnormal uterine bleeding, it would be practically unviable to exclude the presence of endometriosis in one patient completely asymptomatic by laparoscopy, thus, we cannot guarantee that they have no changes in the endometrium. It is a limitation, larger samples are needed for verification in the future. Tregs are the main source of FGL2 in peritoneal fluid of women with endometriosis. CD32B, the receptor of FGL2, was found to be mainly expressed on monocyte and macrophages and showed lower levels of expression on naive T, CD4+ T cells, and ESCs. Previous studies have reported that

FGL2 is upregulated in a number of human immune diseases and malignancies[38]. FGL2 can promote hepatocellular carcinoma xenograft tumor growth and angiogenesis[39], and the overexpression of FGL2 can induce epithelial-to-mesenchymal transition and promote tumor progression in colorectal carcinoma[40], suggesting a tumor-promoting role. Nevertheless, little is understood about the signaling mechanisms mediated by FGL2.

Although FGL2 is highly expressed by Tregs[41], data regarding the specific stimuli required for FGL2 secretion in Tregs are scarce. Locally-produced cytokines are prime candidate molecules. In line with previous work[42], in the present study, we have

## a

- PBS-treated Mφ+naive T
- rFGL2-treated Mφ+naive T

**Th1** — *TBX21* — P=0.0077
relative mRNA expression

**Th17** — *RORC* — P<0.0001
relative mRNA expression

**Th2** — *GATA3* — P<0.0001
relative mRNA expression

**Treg** — *FOXP3* — P=0.0016
relative mRNA expression

## b

- PBS-treated Mφ+naive T
- rFGL2-treated Mφ+naive T

**IFN-γ** — P=0.0037
FCM cytokines (%)

**IL-17A** — P<0.0001
FCM cytokines (%)

**IL-4** — P=0.0135
FCM cytokines (%)

**TGF-β**
FCM cytokines (%)

**Fig. 9 FGL2-polarized pro-repair macrophages induced Th2 and Tregs differentiation.** FGL2-treated macrophages were cocultured with naive T cells at a ratio of 1:4, mRNA and cytokines secretion levels were determined by qRT-PCR (**a**) and FCM (**b**). Data represent the mean value ± SEM of five individual experiments (*n* = 5).

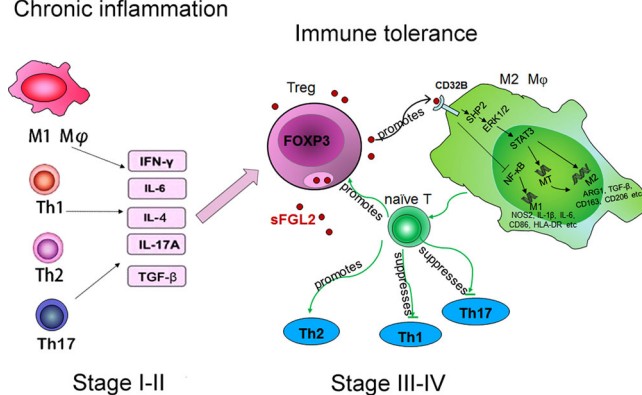

**Fig. 10 Schematic diagram of this study.** In stage I–II endometriosis, the increased secretion of inflammatory cytokines in ectopic milieu induces FGL2 secretion in Tregs. sFGL2 induces pro-repair macrophages skewing through the suppression of NF-κB and the activation of SHP2-ERK1/2-STAT3 pathway. sFGL2 also up-regulates MT expression in macrophages through STAT3 activation which further promotes pro-repair macrophage polarization. In addition, sFGL2-induced pro-repair macrophages inhibit Th1 and Th17 cell differentiation, while promote Th2 and Tregs differentiation. Taken together, sFGL2 mediated cross-talk between Tregs and macrophage alters the balance of pro- and anti-inflammatory microenvironments and facilitates the formation of immune tolerance in stage III-IV endometriosis.

found that IFN-γ, IL-6, IL-4, IL-17A, and TGF-β are responsible for increased FGL2 levels observed in endometriosis. Furthermore, we have explored potential mechanisms of its regulation, and we found potential binding sites for transcription factors NF-κB, STAT1, and C/EBPα in the *FGL2* promoter region. Our quantitative real-time-PCR results reveal the differential pathways that may involve in the cytokine-induced upregulation of FGL2 expression in Tregs.

We elucidate the molecular mechanisms by which FGL2 induces pro-repair macrophages polarization. We used microarray and gene regulation to observe intracellular and intercellular interactions. FGL2-treated macrophages expressed more pro-repair associated genes, such as *CD163*, *IL-10*, *ARG1*, *FGF2*, *TGF-β1*, and *MMP9*, as well as transcription factors. FGL2 was previously found to suppress DCs maturation by interacting with its receptor CD32B expressed on DCs. The macrophages, another important antigen presenting cell[43], also expresses high levels of CD32B. However, little research has been conducted on the effects of FGL2 on macrophages so far. Our data indicate that sFGL2 from Tregs downregulates CD86 and HLA-DR expression on macrophages, implying the potential immunoregulatory effects of adaptive immunity (Tregs) on innate immunity (macrophages). Further, FGL2 may play a role in T cell differentiation through the downregulation of costimulatory molecules on macrophages. More specifically, our data suggest that FGL2 induces pro-repair macrophages polarization by binding to

CD32B, which is consistent with previous reports[17,38]. There are some other markers that can also indicate the activation states of macrophages, the markers used in this study are limited. Pro-repair macrophages, unlike pro-inflammatory macrophages that are pro-inflammatory and cytotoxic, are immunosuppressive and favor angiogenesis and tissue repair[21]. The alteration of the balance between these two subclasses of macrophages may be involved in the pathogenesis of endometriosis[22].

We specifically found that NF-κB suppression and SHP2-ERK1/2-STAT3 activation are associated with the polarization of FGL2-induced macrophages. Macrophages can be driven toward pro-repair phenotypes by canonical pro-repair stimuli such as IL-4, IL-13, and IL-10 by activating STAT6 or STAT3 signal pathways through IL-4Rα or IL-10R[44,45]. NF-κB is a key transcription factor that drives pro-inflammatory macrophages activation and regulates the expression of many inflammatory genes including TNF, IL-1β, IL-6, and IL-12[46,47]. FGL2 may thus be identified as a new molecule that can regulate macrophages polarization.

CD32B is an inhibitory receptor that contains an immuno-receptor tyrosine-based inhibitory motif (ITIM) in its cytoplasmic tail[48]. ITIMs are critical for regulating cellular signal-transduction events. The inhibitory action of ITIM-containing receptors has been linked to the recruitment of proteins such as the SH2-containing tyrosine phosphatases (SHP) SHP1 and SHP2, and the SH2 domain contains inositol phosphatases (SHIP)[35,36]. We have demonstrated here that SHP2 phosphorylation is a positive regulator of ERK1/2-STAT3 pathways. SHP2 mediates the signaling of growth factors and cytokines such as EGF, hepatocyte growth factor, and interleukin-6[49]. In particular, SHP2 is involved in the activation of the ERK1/2 MAP kinase pathway by EGF[50]. It is possible that after recruitment to the phosphorylated ITIM, SHP2 phosphorylates and thereby activates ERK1/2 and then further activates STAT3. It is not known yet whether CD32B ITIM recruitment is the only mechanism for SHP2 phosphorylation, as there may be other pathways of SHP2 activation. Alternatively, SHP2 may perform different roles when recruited to distinct portions of the receptor complex. Therefore, we propose that intracellular roles of the SHP2-ERK1/2 signaling pathway may depend on the molecules and cells involved. In short, FGL2 functions to induce pro-repair macrophages polarization mainly through two mechanisms: the suppression of the NF-κB signaling pathway and the activation of SHP2-ERK1/2-STAT3 signaling pathways.

Interestingly, we found that metallothionein (MT) is strongly upregulated in macrophages treated with FGL2. MT can suppress cytotoxic T lymphocyte and NK cell activity[32,33]. Moreover, MT expression has been shown to be increased in functionally tolerogenic dendritic cells (DCs) and to promote the expansion of Tregs[51]. A previous study also reported the higher expression of the MT2A isoform in IL-10-induced pro-repair macrophages[34], suggesting that MT could be associated with pro-repair macrophages polarization. Indeed, MT is also strongly upregulated in endotoxin-tolerized MDM, which exhibited a pro-repair phenotype[52]. We propose that higher MT levels may further promote macrophages polarization, and we have demonstrated that MT overexpression induces a pro-repair phenotype. Our knowledge is still very limited, and the limitations do not allow reliable predictions of the exact role of MT in macrophages polarization; the biological role of these molecules awaits further investigation.

Our data also show that FGL2-induced pro-repair macrophages promote Th2 and Tregs differentiation, but inhibit Th1 and Th17 differentiation, explaining the shifting of the Th1/Th2 balance toward Th2 responses in endometriosis[53]. Macrophages and T cells can interact with one another to participate in the immune response. IL-4 and IL-13 are the signature cytokines of the Th2-type immune response. IL-4- and/or IL-13-dependent Th2-driven responses can cause pro-repair macrophages not to present antigens to T cells, but to produce minimal amounts of pro-inflammatory cytokines and to act as direct counterparts to Th1-elicited pro-inflammatory macrophages[54]. Tregs can also induce pro-repair macrophages polarization by producing immunosuppressive cytokine IL-10[54]. Therefore, our study further shows that specific interactions occur between macrophages and T cells.

Our current study provides another insight into the understanding of immunological aspects in endometriosis. We propose that during stage I-II endometriosis, pelvic inflammatory cytokines may induce FGL2 production in Tregs. An increased level of FGL2 induces pro-repair macrophages skewing through the suppression of NF-κB and activation of the SHP2-ERK1/2-STAT3 pathway. The FGL2-induced pro-repair macrophages further promote Th2 and Tregs differentiation, which alters the balance between pro- and anti-inflammatory microenvironments in endometriosis and facilitates the formation of immune tolerance in stage III-IV endometriosis. Collectively, these data show that FGL2 secreted by Tregs can function as an important factor in endometriosis progression through direct and indirect regulation of the immune functions of macrophages and T cells. Targeting Tregs/FGL2 may constitute a viable therapeutic approach to treating endometriosis.

## Methods

**Patients and samples.** This study was approved by the Ethics Committees of Obstetrics and Gynecology Hospital of Fudan University, and informed consent was obtained from all subjects. In total, 48 endometriosis patients (mean age 33.4 ± 4.7 years) were surgically staged, including 27 in stage I–II, and 21 in stage III–IV according to the revised American Fertility Society classification. Exclusion criteria comprised presence of other pelvic pathologies and having used hormone therapies in the 3 months preceding surgery. The enrolled patients were submitted to laparoscopic surgery and curettage. During surgery, ectopic (ovarian endometriosis) and matched eutopic endometrium was collected together with peritoneal fluid (2–10 ml). Normal endometrial tissues were collected from 20 age-matched patients who underwent curettage and who were not found to have any pathological lesions. Controlled peritoneal fluid was collected from 34 women who underwent laparoscopy for cystic teratoma or hysteromyomectomy. The surgeries were performed during the proliferative phase of the menstrual cycle. Heparinized peripheral venous blood samples for the isolation of human PBMCs were collected from healthy volunteers. All samples were collected under sterile conditions and were processed for further analysis.

**Cell isolation, culture, and treatment.** Primary culture of endometrial stromal cells (ESCs) and endometrial epithelial cells (EECs) from healthy control patients was performed by the mechanical dispersal method. Briefly, tissue fragments were washed and cut into small fragments with scissors. After being digested in DMEM/F12 (Gibco, CA, USA) containing type IV collagenase (Sigma, MO, USA) (1 mg per ml) for 35 min at 37 °C, dispersed cells were filtered through a 400-mesh sieve. Cell filtrates were collected, centrifuged, and resuspended in DMEM/F-12 containing 10% FBS. Enriched ESCs were obtained by plastic adhesion in cell culture flasks. To obtain endometrial epithelial cells (EECs), backwashed cells drawn from a 400-mesh sieve were collected and cultured.

PBMC were isolated from heparinized peripheral venous blood samples via Ficoll density gradient centrifugation. The peritoneal fluid was collected and centrifuged for isolating the immune cells. CD14 MicroBeads (Miltenyi Biotec, Germany) were used for the magnetic positive selection of human CD14$^+$ monocyte/macrophage from PBMC and the cells from peritoneal fluid. Monocyte were further differentiated into macrophages (monocyte-derived macrophages, MDM) by incubation with M-CSF (50 ng per ml, Peprotech, USA) for 6 days. Macrophages were polarized to pro-inflammatory macrophages by incubation with 20 ng per ml of IFN-γ (Peprotech, USA) and 10 ng per ml of LPS (Sigma, MO, USA). Pro-repair macrophages polarization was performed by incubation with 20 ng per ml of interleukin 4 (IL-4) (Peprotech, USA). THP-1 and U937 monocytic cell lines were purchased from ATCC and cultured in RPMI containing 10% FBS. For differentiating into macrophages, THP-1 and U937 cells were incubated with 100 ng per ml phorbol 12-myristate 13-acetate (PMA) for 48 h. NK cells were isolated and purified by MACS negative selection. For isolation of NK cells, the isolated PBMC or the cells from peritoneal fluid were incubated successively with a cocktail of biotin-conjugated antibodies and the NK microbeads cocktail. A cell suspension was placed onto a MS column (Miltenyi Biotec, Germany), which was placed in the magnetic field of a MACS separator. Flow through-containing

unlabeled cells was then collected, representing the enriched NK. Isolation of highly pure NK was achieved through the depletion of magnetically labeled cells.

CD4⁺CD25⁺ Tregs were purified using the CD4⁺CD25⁺ Regulatory T Cell Isolation Kit (Miltenyi Biotec, Germany). Briefly, isolation was performed in a two-step procedure. First, non-CD4⁺ T cells were indirectly magnetically labeled with a cocktail of biotin-conjugated antibodies and Anti-Biotin MicroBeads. The labeled cells were subsequently depleted. In the second step, the flow-through fraction of the pre-enriched CD4⁺ T cells was labeled with CD25 MicroBeads for the positive selection of CD4⁺CD25⁺ regulatory T cells. The isolated Tregs were subsequently expanded using the Tregs expansion kit (Miltenyi Biotec, Germany) according to the manufacturer's protocols. The isolation of naive CD4⁺ T cells was performed using a naive CD4⁺ T cell isolation kit II (Miltenyi Biotec, Germany) for magnetic negative selection. When necessary, rFGL2 (1 µg per ml, Abnova) or the same volume of PBS (as control) was added. SHP2 inhibitor (SHPi, 100 µM, Millipore, USA), ERK inhibitor (ERKi, 10 µM, Selleck, USA), and STAT3 inhibitor (STAT3i, 1 µM, Selleck, USA) were added to treat THP-1 and U937 cells for 24 h. FGL2-treated macrophages were cocultured with naive T cells at a ratio of 1:4.

**Enzyme-linked immunosorbent assay (ELISA).** Peritoneal fluid samples and cell culture supernatant were centrifuged to remove cellular debris and were stored at −80 °C until analyzed. The concentration of FGL2, IL-12p70, and IL-10 were measured by using human FGL2, IL-12p70, and IL-10 ELISA Kits (BioLegend, CA, USA, and R&D system, USA) according to the manufacturer's instructions.

**Immunohistochemistry and immunofluorescence assay.** Briefly, for the immunohistochemical assay, formalin-fixed, paraffin-embedded tissue from all samples was sliced into 4 µm sections. After deparaffinization and rehydration, antigen retrieval was performed using the heat-induced epitope retrieval method with citrate buffer. After incubating with the primary antibodies (anti-FGL2, dilution 1:50, Abcam; anti-CD32B, dilution 1:100, Abcam) at 4 °C overnight, the sections were washed and then incubated with the secondary antibody for 30 min and with horseradish peroxidase for another 15 min before adding the DAB Chromogen (ZSGB-Bio). The slides were counterstained with hematoxylin and subsequently dehydrated and mounted. The relative expression of FGL2 and CD32B in each slide was analyzed with Image Pro Plus 6.0 software. For the immunofluorescence assay, the following primary antibodies were used: anti-CD32B goat antibody (dilution 1:100, Abcam, MA, USA) or anti-CD32B rabbit antibody (dilution 1:100, Abcam, MA, USA), anti-CD68 mouse antibody (dilution 1:100, Abcam, MA, USA), anti-NCAM1 (RNL-1, CD56) mouse antibody (dilution 1:100, Abcam, MA, USA), and anti-CD4 mouse antibody (15 µg per ml, R&D, USA). The second antibodies used were donkey anti-goat IgG (Alexa Fluor 488, 1:250, Abcam, MA, USA), donkey anti-mouse IgG (Alexa Fluor 647, 1:250, Abcam, MA, USA), or goat anti-rabbit IgG (Alexa Fluor 488, 1:200, Abcam, MA, USA), and goat anti-mouse IgG (Alexa Fluor 647, 1:200, Abcam MA, USA). After DAPI staining (nuclear, 1 µg per ml, CST, MA), images were acquired using a Leika TCS SP5 confocal microscope.

**Flow cytometry.** Cells were stained with antibodies to cell surface antigens for 30 min in the dark on ice. For intracellular cytokine staining, the cells were stimulated with PMA (50 ng per ml, Sigma, USA), ionomycin (1 µg per ml, Sigma, USA), and monensin (1 µl per ml, BD Biosciences, USA) 4 h before testing. Intracellular staining procedure was carried out according to the manufacturer's instructions with the Fix & Perm cell permeabilization kit (BioLegend, CA, USA). Flow cytometry was performed on a CyAn ADP machine (Beckman coulter, USA) and analyzed with Flowjo software (Tree Star). Antibodies used for flow cytometry were purchased from BioLegend, eBioscience, and BD Biosciences.

**Quantitative real-time reverse-transcriptase polymerase chain reaction (qRT-PCR).** Gene expression was determined by qRT-PCR analysis using the GAPDH gene as a housekeeping gene. The total RNA was isolated according to a standard TRIzol RNA isolation protocol (Invitrogen). Total RNA (1 µg) was used in 20 µl of a reverse transcription reaction mixture (Takara) for synthesizing cDNA. A real-time PCR analysis was then performed using the SYBR Fast qPCR Mix Kit (Takara). Real-time PCR amplification and detection were performed using 384-well reaction plates and the Applied Biosystems 7900 Fast Real-time PCR system. Relative fold induction levels were calculated using the comparative CT method for separate tube amplification. Primers used were designed and purchased from Sangon Biotech and are listed in Supplementary Table 1.

**Gene expression microarray analysis.** RNA samples were obtained after the treatment of MDM (4 different healthy donors) with rFGL2 using GeneChip Human Transcriptome Array 2.0 (HTA 2.0, Affymetrix). Microarray data were analyzed using the Gene-Cloud of Biotechnology Information (GCBI) platform. To identify genes affected by rFGL2 by Gene Ontology analysis, genes were selected with a minimum fold change of 1.5 and based on an adjusted P value < 0.05.

**Expression vectors and SiRNA transfection.** Cells were routinely grown in medium with 10% FBS and seeded at a density of 1 × 10⁵ cells per well in 6-well plates. For expression vector transfection, when cells were grown to 90% confluence, they were switched to serum-free medium 2 h prior to transfection. Plasmid DNA (4 µg) was incubated with 8 µl of Lipofectamine 2000 (Invitrogen, USA) and then was added to the cells. To transfect siRNA, the cells were grown to 50% confluence, and for each transfection reaction, 100 pmol of siRNA was incubated with 5 µl of Lipofectamine 2000. The medium was changed to complete medium with 10% FBS after 5 h, and the cells were used for further study at the indicated time.

**Western blot.** Total protein was extracted from cultured cells using lysis buffer, and protein concentrations were determined by using BCA protein assay (Beyotime, Shanghai, China). Total protein (30 µg) was loaded per lane. Primary antibodies were probed against pNF-κB and NF-κB (Cell Signaling, 1:500); pERK1/2 and ERK1/2 (Cell Signaling, 1:500); pSTAT3 and STAT3 (Cell Signaling, 1:500); pSHP2 and SHP2 (Cell Signaling, 1:500) at 4 °C overnight. After further incubation with the second antibodies (anti-Mouse or anti-Rabbit, Bioworld, 1:10000) for 1 h at room temperature, specific proteins were visualized with ECL (Millipore) on the ImageQuant LAS 4000 system (GE Healthcare, USA).

**Statistics and reproducibility.** Statistical tests were performed using GraphPad Prism 8.0 software. Student's t tests were used to determine the significance between two groups and ANOVA tests were applied for multiple comparisons. Data are expressed as the mean ± SEM values. P values are denoted in the figures. P values of <0.05 were considered statistically significant. The samples are sufficient for ELISA assay of peritoneal fluid, IHC, and IF analysis. ELISA for cell supernatant, FCM and RT-PCR were set up in duplicates or triplicates and tested in n = 3 or 5 independent experiments (as indicated in each Figure). Western blot was assayed in n = 3 independent experiments.

**Reporting summary.** Further information on research design is available in the Nature Research Reporting Summary linked to this article.

## Data availability

The accession number for the Gene Expression Array data reported in this paper is GEO: GSE133867. All other data needed to evaluate the conclusions in the paper are present in the paper or the Supplementary Information. All source data generated and analyzed during the current study are available in Supplementary Data 1. Data can also be obtained from the corresponding author upon reasonable request.

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

## Acknowledgements

This work was supported by grants from the Ministry of Science and Technology of China (2017YFC1001400 and 2015CB943300 to Da-Jin Li); the National Natural Science Foundation of China (item numbers 81971456 and 81200425 to Xiao-Qiu Wang); and the National Natural Science Foundation of China (item numbers 81471548 and 81490744 to Da-Jin Li).

## Author contributions

Xin-Xin Hou and Xiao-Qiu Wang conducted all the experiments, prepared the figures, and wrote the manuscript. Wen-Jie Zhou assisted with cell culture, examined the patients, collected specimens, and generated clinical data. Da-Jin Li conceived of and designed the study, oversaw the completion of the study, and edited and finalized the manuscript.

## Competing interests

The authors declare no competing interests.
