## [Peer Review File · Communications Biology]

Reviewers' comments:

Reviewer #1 (Remarks to the Author):

First of all, I would like to congratulate the authors for their work. Overall, very well written, conducted in detail, and careful in the analyzes and interpretations. The theme is very well situated in the introduction, it is current, and both the gap and the hypothesis of the authors is plausible. Figures have a very good quality.

\$ major comments

- Introduction: the authors cited in lines 98-100 just one sentence about M1-M2 polarization and involvement in diseases and inflammatory conditions. I believe that the information already in the literature specifically regarding the M1 / M2 macrophage subtypes ratio in endometriosis should be better described. Recently, two interesting studies has been published about this issue. A meta-analysis (doi: 10.1038 / s41598-019-57207-y) that used the transcriptome of eutopic endometrium to predict the cell types in microenvironment which has shown that a predominance of M2 macrophage in relation to M1 subtype seems to be pivotal in endometriosis severity independently of the hormonal milieu; and a experimental study (10.1016 / j.fertnstert.2019.08.060) that evaluated RNA-seq of M1 / M2 macrophages isolated from eutopic endometrium of women with endometriosis by FACS. Adding and detailing these informations could be useful for authors and for readers.
- Material and methods: Why were control patients undergone to curettage? Did they have any pathology or indication for the procedure?
- Author inform that surgeries were performed in proliferative phase. How do they define this cycle phase?

\$ minor comments

- in line 66, please use the full description of fibrinogen-like protein 2 (FGL2) before its abbreviation, as well as sFGL2 (FGL2 soluble form). I think that as it plays a central role in the manuscript, it should be described.
- in lines 107-110, the sentence "Our data demonstrate that high levels of sFGL2 secreted by Tregs in endometriosis may shift macrophages polarization to the M2 / anti-inflammatory phenotype, which may accelerate the progression of endometriosis" is not adequate to the section . It is a synthesis of the results, because of that should be in discussion, not in introduction.
- in line 113, I suggest deleting the word "production". In fact, the authors did not measure production, they have evaluated the levels.
- in line 119 the "... to be positively correlated to disease progression" does not seem to me the most correct one. It sounds interpretative. Besides that, the authors did not inform any correlation analysis in materials and methods. I would prefer the statement that the levels are higher in III-IV endometriosis
- sentences in lines 119-123, 127-128 seem to me interpretative too. As I have said before, in my opinion, they should be in discussion.

Reviewer #2 (Remarks to the Author):

The authors present a compelling study showing a role for tregs in producing sFGL2 that effectively induces macrophages to polarize such that they exhibit a pro-repair phenotype. They also demonstrate the signalling pathway by which pro-repair genes are expressed following stimulation with FGL2.

The authors present sound experimental methods that shed new light on potential pathogenic mechanisms in endometriosis but I do have some concerns over choice of sample type for some experiments which may mean that some conclusions are not fully supported. Methodology described sufficiently such that the work could be reproduced, stats look fine.

Main points:

- The M1 - M2 classification is too outdated and the authors need to put more emphasis on the huge spectrum of macrophage phenotype. It is preferable to refer to macs as pro-repair or pro-inflammatory. Please remove all reference to M1 and M2 from paper. The authors rely heavily on in vitro polarization of PBMCs using prototypical pro-repair and pro-inflammatory cytokines. It is well documented that these in vitro phenotypes do not represent the in vivo situation where phenotype is more complex. Moreover the markers used are limited because co-stimulatory molecules are not reliable pro-inflam markers. This is a limitation of the study.
- It is not clear in the methods whether the stromal and epithelial cells are from healthy patients or women with endometriosis. It is important to clarify.
- The isolated immune cells for analysis of FGL2 levels are from peripheral blood. It would be more relevant to look at individual immune cell populations from peritoneal fluid and compare endometriosis to no endometriosis.

Minor points:

- The opening sentence of the abstract does not make sense and requires some re-wording.
- Introduction - please include some info on MT. It seemed to come a bit left field in the results.
- Fig1 scale bars and negatives please!
- Line 136-137 - state whether these cells are primary, and whether they are from women with or without endometriosis.
- Line 137 - state origin of immune cells.
- Fig3 - the direct link of transcription factor to FGL2 expression is not shown - the language should be toned down here. e.g. we speculate the mechanism....
- Methods - more details on the subtype of lesions included in the study please (e.g. IHC analysis).

Reviewers' comments:

Reviewer #1 (Remarks to the Author):

First of all, I would like to congratulate the authors for their work. Overall, very well written, conducted in detail, and careful in the analyzes and interpretations. The theme is very well situated in the introduction, it is current, and both the gap and the hypothesis of the authors is plausible. Figures have a very good quality.

\$ major comments

- Introduction: the authors cited in lines 98-100 just one sentence about M1-M2 polarization and involvement in diseases and inflammatory conditions. I believe that the information already in the literature specifically regarding the M1 /M2 macrophage subtypes ratio in endometriosis should be better described. Recently, two interesting studies has been published about this issue. A meta-analysis (doi: 10.1038 / s41598-019-57207-y) that used the transcriptome of eutopic endometrium to predict the cell types in microenvironment which has shown that a predominance of M2 macrophage in relation to M1 subtype seems to be pivotal in endometriosis severity independently of the hormonal milieu; and a experimental study (10.1016 / j.fertnstert.2019.08.060) that evaluated RNA-seq of M1 / M2 macrophages isolated from eutopic endometrium of women with endometriosis by FACS. Adding and detailing these informations could be useful for authors and for readers.

Response: It is indeed not sufficient to describe the M1-M2 polarization and involvement in diseases and inflammatory conditions in the introduction, thank you very much for providing these useful articles, we have added more information and cited these two interesting studies in the revision.

- Material and methods: Why were control patients undergone to curettage? Did they have any pathology or indication for the procedure?

Response: The control patients were suffering from irregular uterine bleeding (such as prolonged menstrual bleeding, etc.), or have diverticulum after cesarean section, before undergoing further therapy, they regularly received curettage to exclude endometrial diseases.

- Author inform that surgeries were performed in proliferative phase. How do they define this cycle phase?

Response: The surgeries were performed 3-7 days after menstrual bleeding stops, thus this period is the proliferative phase. Moreover, the pathology report can also help us judge the phase of the menstrual cycle.

\$ minor comments

- in line 66, please use the full description of fibrinogen-like protein 2 (FGL2) before its abbreviation, as well as sFGL2 (FGL2 soluble form). I think that as it plays a central role in the manuscript, it should be described.

Response: Yes, it is our oversight, we have added the full description according to your kind suggestion.

- in lines 107-110, the sentence “Our data demonstrate that high levels of sFGL2 secreted by Tregs in endometriosis may shift macrophages polarization to the M2 / anti-inflammatory phenotype, which may accelerate the progression of endometriosis” is not adequate to the section. It is a synthesis of the results, because of that should be in discussion, not in introduction.

Response: We have re-written this sentence according to your good suggestion.

- in line 113, I suggest deleting the word “production”. In fact, the authors did not measure production, they have evaluated the levels.

Response: Yes, we have evaluated FGL2 levels, not its production, we have deleted the word “production” according to your kind suggestion.

- in line 119 the “... to be positively correlated to disease progression” does not seem to me the most correct one. It sounds interpretative. Besides that, the authors did not inform any correlation analysis in materials and methods. I would prefer the statement that the levels are higher in III-IV endometriosis

Response: We have revised it according to your kind suggestion.

- sentences in lines 119-123, 127-128 seem to me interpretative too. As I have said before, in my opinion, they should be in discussion.

Response: We have done the revision as you suggested.

Reviewer #2 (Remarks to the Author):

The authors present a compelling study showing a role for tregs in producing sFGL2 that effectively induces macrophages to polarize such that they exhibit a pro-repair phenotype. They also demonstrate the signalling pathway by which pro-repair genes are expressed following stimulation with FGL2.

The authors present sound experimental methods that shed new light on potential

pathogenic mechanisms in endometriosis but I do have some concerns over choice of sample type for some experiments which may mean that some conclusions are not fully supported. Methodology described sufficiently such that the work could be reproduced, stats look fine.

Main points:

- The M1 - M2 classification is too outdated and the authors need to put more emphasis on the huge spectrum of macrophage phenotype. It is preferable to refer to macs as pro-repair or pro-inflammatory. Please remove all reference to M1 and M2 from paper. The authors rely heavily on in vitro polarization of PBMCs using prototypical pro-repair and pro-inflammatory cytokines. It is well documented that these in vitro phenotypes do not represent the in vivo situation where phenotype is more complex. Moreover the markers used are limited because co-stimulatory molecules are not reliable pro-inflam markers. This is a limitation of the study.

Response: Indeed, functional polarization of macrophages into only two groups is an over-simplified description of macrophage heterogeneity and plasticity. We have revised it, referred to macs as pro-repair or pro-inflammatory in the manuscript as you suggested.

There are some useful markers are widely recognized as characteristics of pro-inflammatory and pro-repair macrophages. Such as IL-10, TGF- β , CD206 and Arg-1 represent pro-repair phenotype, while IL-6, IL-12p70 and NOS2 represent pro-inflammatory phenotype, we have detected these markers in this study.

Indeed, there are some other markers that can also indicate the activation states of macrophages, the markers used in this study are limited, this is a limitation, we added this in the discussion and we will pay more attention to more comprehensive set of markers in our future work. Thanks for your kind suggestion.

- It is not clear in the methods whether the stromal and epithelial cells are from healthy patients or women with endometriosis. It is important to clarify.

Response: The stromal and epithelial cells were isolated from healthy patients, we have added the information in the methods.

- The isolated immune cells for analysis of FGL2 levels are from peripheral blood. It would be more relevant to look at individual immune cell populations from peritoneal fluid and compare endometriosis to no endometriosis.

Response: This is really important. Thanks for your constructive suggestion. We have detected FGL2 levels of individual immune cell populations from peritoneal fluid and compared its levels between women with and without endometriosis. We found that Treg in peritoneal fluid of patients with endometriosis secreted significantly higher levels of FGL2 than that of the control. The supplementary data are shown in Figure 2.

Minor points:

- The opening sentence of the abstract does not make sense and requires some re-wording.

Response: We have rewritten the opening sentence of the abstract as you suggested.

- Introduction - please include some info on MT. It seemed to come a bit left field in the results.

Response: We have included the information on MT in the introduction as you suggested.

- Fig1 scale bars and negatives please!

Response: We have added the scale bars in Figure 1 as you suggested, the isotype images are served as negative controls.

- Line 136-137 - state whether these cells are primary, and whether they are from women with or without endometriosis.

Response: We have added the information according to your kind suggestion.

- Line 137 - state origin of immune cells.

Response: We have done it as you suggested.

- Fig3 - the direct link of transcription factor to FGL2 expression is not shown - the language should be toned down here. e.g. we speculate the mechanism....

Response: Yes, we have revised the wording according to your kind suggestion.

- Methods - more details on the subtype of lesions included in the study please (e.g. IHC analysis).

Response: We feel sorry we may not understand exactly what you mean. The lesions were the proliferative phase of the menstrual cycle, we have stated in the study. Would you like to give some hints? Thank you so much.

REVIEWERS' COMMENTS:

Reviewer #1 (Remarks to the Author):

Dear authors,

I congratulate you again for the effort and believe that the manuscript is suitable for publication. Considering your notes on controls, I think it would be worthwhile to insert information in the discussion about this limitation inherent to the case series, but that does not compromise the originality of the study. As the "control" patients had symptoms of abnormal uterine bleeding, it is not possible to guarantee that they have no changes in the eutopic endometrium, but on the other hand, it was possible to exclude the presence of endometriosis by laparoscopy, which would be practically unviable in one patient. completely asymptomatic.

Reviewer #2 (Remarks to the Author):

The authors have made a considerable effort with their manuscript revision, however there are still some points that require attention.

1. Intro, line 119. MT are still not defined despite their function being described.
2. Results. Thanks for adding in the PF data. Please also show flow plots for the different immune cells for peripheral blood and PF.
3. Discussion - take out 'we will pay more attention next time ...' it undermines the study.
4. Methods - please state what subtypes of lesions were included in the study. i.e. superficial peritoneal, ovarian endometrioma or deep infiltrating.
5. There are some points in the response to reviewer 1 where it is not clear whether the authors have clarified information in the manuscript. If the reviewer asks a questions then it is because the information is missing from the manuscript. It is helpful when responding to reviews to state exactly where in the manuscript you have added the extra information i.e. line number.

Reviewer #1 (Remarks to the Author):

Dear authors,

I congratulate you again for the effort and believe that the manuscript is suitable for publication. Considering your notes on controls, I think it would be worthwhile to insert information in the discussion about this limitation inherent to the case series, but that does not compromise the originality of the study. As the "control" patients had symptoms of abnormal uterine bleeding, it is not possible to guarantee that they have no changes in the eutopic endometrium, but on the other hand, it was possible to exclude the presence of endometriosis by laparoscopy, which would be practically unviable in one patient. completely asymptomatic.

Response: Many thanks for your kind reminding, we have inserted the information about this limitation in the discussion as you suggested (Line 364-370).

Reviewer #2 (Remarks to the Author):

1. Intro, line 119. MT are still not defined despite their function being described.

Response: We are sorry for this omission, we have added more information about MT including the definition in the introduction (Line 119-130).

2. Results. Thanks for adding in the PF data. Please also show flow plots for the different immune cells for peripheral blood and PF.

Response: The different immune cells in peripheral blood and PF were first isolated using MACS with an average purity of more than 90%, then sFGL2 levels of these cells were detected by ELISA assay. We have changed the figures to dot plots.

3. Discussion - take out 'we will pay more attention next time ...' it undermines the study.

Response: Thanks for your kind suggestion. We have deleted this sentence.

4. Methods - please state what subtypes of lesions were included in the study. i.e. superficial peritoneal, ovarian endometrioma or deep infiltrating.

Response: The lesions in this study were all ovarian endometriosis, we have added the information in the results and methods part as you suggested (Line 155, 157-158, 180, 501).

5. There are some points in the response to reviewer 1 where it is not clear whether the authors have clarified information in the manuscript. If the reviewer asks a questions then it is because the information is missing from the manuscript. It is

helpful when responding to reviews to state exactly where in the manuscript you have added the extra information i.e. line number.

Response: Thanks for your information, it is very helpful. We will follow your suggestion. Thanks again.